# Efficacy and Safety of High-Dose Immunoglobulin-Based Regimen in Statin-Associated Autoimmune Myopathy: A Multi-Center and Multi-Disciplinary Retrospective Study

**DOI:** 10.3390/jcm9113454

**Published:** 2020-10-27

**Authors:** Elena Treppo, Maria Infantino, Maurizio Benucci, Viviana Ravagnani, Boaz Palterer, Martina Fabris, Paola Tomietto, Mariangela Manfredi, Maria Grazia Giudizi, Francesca Ligobbi, Daniele Cammelli, Marina Grandis, Paola Parronchi, Salvatore De Vita, Luca Quartuccio

**Affiliations:** 1Department of Medicine, Rheumatology Clinic, University of Udine, ASUFC Udine, 33100 Udine, Italy; treppo.elena@gmail.com (E.T.); quarto77@gmail.com (S.D.V.); 2Allergy and Immunology Laboratory Unit, Department of Laboratory Medicine, San Giovanni di Dio Hospital, 50125 Florence, Italy; maria2.infantino@uslcentro.toscana.it (M.I.); mariangela.manfredi@uslcentro.toscana.it (M.M.); 3Rheumatology Unit, Department of Medicine, San Giovanni di Dio Hospital, 50125 Florence, Italy; maurizio.benucci@uslcentro.toscana.it (M.B.); Francesca.ligobbi@uslcentro.toscana.it (F.L.); 4Rheumatology Unit, Centro Day Hospital Allergologia e Immunologia Clinica, ASST Mantova, 46100 Mantova, Italy; viviana.ravagnani@gmail.com; 5Department of Experimental and Clinical Medicine, University of Florence, 50121 Florence, Italy; boaz.palterer@gmail.com (B.P.); mariagrazia.giudizi@unifi.it (M.G.G.); daniele.cammelli@unifi.it (D.C.); paola.parronchi@unifi.it (P.P.); 6Department of Laboratory Medicine, ASU FC Udine, 33100 Udine, Italy; martina.fabris@asufc.sanita.fvg.it; 7Rheumatology Unit, University of Trieste, 34123 Trieste, Italy; paola.tomietto@asuits.sanita.fvg.it; 8Department of Neuroscience (DINOGMI), San Martino Hospital, IRCCS, 16131 Genova, Italy; mgrandis@neurologia.unige.it

**Keywords:** anti-HMGCR antibody, autoimmune myopathy, immune-mediated necrotizing myopathy, statins

## Abstract

Statin-associated autoimmune myopathy is a rare muscle disorder, characterized by autoantibodies against HMGCR. The anti-HMGCR myopathy persists after statin, and often requires immunosuppressive therapy. However, there is not a standardized therapeutic approach. The purpose of this study is to report the effectiveness of the immunosuppressive treatment employed in a multi-center and multi-disciplinary cohort of patients affected by anti-HMGCR myopathy, in which an immunoglobulin (IVIG)-based treatment strategy was applied. We collected 16 consecutive patients with a diagnosis of anti-HMGCR myopathy, between 2012 and 2019, and recorded data on clinical and laboratory presentation (i.e., muscle strength, serum CK levels, and anti-HMGCR antibody titer) and treatment strategies. Our results highlight the safety and efficacy of an induction therapy combining IVIG with GCs and/or methotrexate to achieve persistent remission of the disease and steroid-free maintenance. Under IVIG-based regimens, clinical improvement and CK normalization occurred in more than two thirds of patients by six months. Relapse rate was low (3/16) and 2/3 relapses occurred after treatment suspension. Nearly 90% of the patients who successfully discontinued GCs were treated with a triple immunosuppressive regimen. In conclusion, an IVIG-based regimen, which particularly includes high-dose immunoglobulin, GCs and methotrexate, can provide a fast remission achievement with GC saving.

## 1. Introduction

Statin-associated autoimmune myopathy is a relatively newly described disorder and the presence of autoantibodies against 3-hydroxy-3-methylglutaryl coenzyme A reductase (HMGCR) have only recently been identified [1,2]. Anti-HMGCR myopathy is a rare side effect of statin therapy, even though it can also develop in statin-naïve patients [3,4,5]. Its incidence is not well defined, though it is estimated in approximately 2–3 of every 100,000 statin-treated patients [6,7]. It usually affects middle-aged people with some cardiovascular risk factors, such as type 2 diabetes, hypertension or hypercholesterolemia, which justify the introduction of statins [8]. The myopathy is characterized by symmetric muscle weakness, markedly elevated serum creatine kinase (CK) levels, abnormal electromyography (EMG) [9], and histologic evidence of muscle cell necrosis and degeneration, along with a lack of significant inflammatory infiltrates and circulating autoantibodies against 3-hydroxy-3-methylglutaryl coenzyme A reductase (HMGCR) [7,10], which have been recognized as possible pathogenic autoantibodies [10,11]. Unlike other immune-mediated myopathies, there are few reports of extra-muscular involvement. Nevertheless, non-specific systemic and extra-muscular symptoms, such as arthritis, Raynaud’s phenomenon and rash, are uncommon [3]. A predisposing genetic background has been also documented, since anti-HMGCR myopathy has one of the strongest associations between an immunogenetic risk factor and autoimmune disease, in particular with the class II human leukocyte antigen (HLA) allele D related B (DRB)1*11:01 [12]. Despite statin discontinuation, anti-HMGCR myopathy can persist and require long-term immunosuppressive therapy [13]. The age at onset can influence the outcome, with younger patients having more severe muscle disease than older patients and a worse prognosis [14]. There are no guidelines for therapy, to date; however, some studies suggest that high-dose intravenous immunoglobulins (IVIG) could be a promising therapy, consistently with their efficacy in other autoimmune myositis [15], while glucocorticoids (GCs) may not be the cornerstone of this disease, differently from polymyositis [13].

The aim of the present study is to provide a further support to the prompted use of immunosuppressive IVIG-based treatment in anti-HMGCR myopathy. To this end, a retrospective analysis of a multi-center cohort of patients suffering from anti-HMGCR myopathy was carried out. Importantly, the patients were followed by different medical specialists, all of them concordant, aiming to reach a complete disease remission after induction therapy as a target of treatment, and a minimal use or avoidance of GCs during the following maintenance therapy. During the induction phase, all the participating centers used IVIG with or without methotrexate (MTX), in addition to GCs. The results reported here support the efficacy of the IVIG-based regimens, and, in particular, a triple-therapy induction strategy (high-dose IVIG, MTX, and GCs) for the anti-HMGCR myopathy. Moreover, this IVIG-based regimen minimized the use of GCs, in association with a steroid-sparing immunosuppressant (SSI), during the maintenance phase.

## 2. Experimental Section

### 2.1. Data Collection

This is a retrospective study based on electronic clinical chart records. Consecutive patients suffering from statin-associated autoimmune myopathy were collected in six Italian specialized centers between 2012 and 2019, from different specialties, i.e., Rheumatology, Allergy and Clinical Immunology, and Neurology. Data on demographics, statin use, myopathic features (i.e., clinical manifestations, serum CK levels, anti-HMGCR antibody at onset), and treatment strategies were collected. The Medical Research Council of Great Britain (MRC) muscle strength grading system was used to perform manual muscle testing (MMT) [16]. Clinical remission of disease was defined as the absence of muscle symptoms, recovered muscle strength based on MRC score evaluation, and normalization of serum CK levels. Clinical relapse was defined as the reappearance of muscle symptoms and worsening of muscle strength, and raise in serum CK levels beyond the upper limit defined by reference laboratories. The induction therapy was defined as the first chosen treatment from diagnosis to clinical remission. IVIG was given at the dose of 0.4 g/kg daily for 5 days, monthly, for at least 3 months. The initial daily dose of GCs corresponded to 1 mg/kg prednisone-equivalent. Maintenance therapy was defined as the treatment that followed the achievement of remission.

### 2.2. Eligibility Criteria

The diagnosis of anti-HMGCR myopathy in this retrospective study required the following criteria:-Clinical and laboratory evidence of muscle damage, including objective decrease in muscle strength by using MMT, elevation in serum CK levels, myopathy pattern in EMG and/or muscle biopsy);-Positive anti-HMGCR antibody (cut-off value >20 UA/mL) measured by chemiluminescence assay (BioFlash, Inova, CA, USA).

### 2.3. Statistical Analysis

Descriptive statistics are used to summarize the baseline characteristics of the study. Continuous data are reported as mean with standard deviation (SD), or median with range. Categorical data are reported as counts with percentages.

### 2.4. Compliance with Ethical Standards

The authors assert that all the procedures contributing to this work comply with the ethical standards of the relevant national and institutional committees on human experimentation and the Helsinki Declaration of 1975, as revised in 2008. This article does not contain any studies of human or animal subjects performed by any of the authors. The retrospective study was approved by the Local Institutional Review Boards (IRB) from all the centres involved (University of Udine IRB, coordinating centre, prot. n. 79,577). The off-label use of IVIG for anti-HMGCR myopathy was approved by each hospitals. All the patients were asked to sign an informed consent for accessing their medical records and using their clinical data for research purposes and improvement in healthcare. The advice of the adverse event under statin use was sent to the Italian Agency for the Drug.

## 3. Results

### 3.1. Clinical Features

Between 2012 and 2019, a total of 16 consecutive patients (seven females, nine males) with anti-HMGCR myopathy were collected. The mean age at onset of myopathy was 72.4 ± 10.3 years. All the patients (16/16) were Caucasian, 7/16 (43.8%) had type 2 diabetes, 8/16 (50%) had cardiovascular disease, 11/16 (68.8%) had arterial hypertension, and none had cancer at disease onset and during the follow-up. Thirteen out of 16 patients (81.3%) had been exposed to statin, of whom 1/13 (7.7%) were exposed to red yeast rice, while 3/16 (18.7%) were not exposed. All the patients had significant muscle symptoms (muscle weakness and/or myalgia) and positive anti-HMGCR antibodies at diagnosis. The median (range) MRC score was 3 (1 to 4) (Table 1). The median (range) serum CK level at the onset of the disease was 5691 IU/L (359-13171). Seven out of 16 (43.8%) patients showed oropharyngeal dysphagia, with one of them requiring parenteral nutrition. The EMG showed a myopathy pattern in 14/16 (87.5%) patients. The muscle biopsy was performed in 10/16 (62.5%) patients, with a documented necrotizing myopathy in nine (90%) and inflammatory cell infiltrates in one (1/10, 10%). The baseline clinical characteristics of the patients are reported in Table 2. The characteristics of each patient together with some details regarding clinical presentation, CK levels at onset, histopathological features and treatment outcome are reported in Table 1.

### 3.2. Treatment and Outcome

As induction therapy, 11/16 (68.8%) patients received a triple induction therapy constituting of the combination of high-dose IVIG, MTX and GCs. Two out of 16 (12.5%) patients were treated with a double induction therapy with high-dose IVIG and GCs without MTX, and 2/16 (12.5%) with GCs alone because of contraindications (ongoing polypharmacy, elderly patients), based on physician’s choice. Finally, the only patient exposed to red yeast rice had red yeast rice suspension only. Clinical remission and normalization of CK levels within month +24 were obtained in all the patients. Gradual improvement started soon after the first month, and nine patients among the 13 treated with the double or triple therapy (69.2%) completely recovered and normalized the CK value within 6 months. More details about induction and maintenance therapy are depicted in Figure 1.

The maintenance therapy was based on SSI, with or without IVIG, and GCs. As previously defined, maintenance therapy started at remission achievement and was tailored to each patient. It was characterized by a progressive tapering of GCs (complete GCs-discontinuation achieved in 9/16 (56.3%)), the use of SSI (mainly MTX), and occasionally the administration of IVIG cycles. The IVIG-maintenance therapy schedule was based on the physician’s choice, and it occurred periodically or on demand. At the last visit, 9/16 (56.3%) patients were receiving SSI (7/9 (77.8%) MTX, 1/9 (11.1%) azathioprine (AZA), 1/9 (11.1%) rituximab (RTX)), in combination with low doses of GCs in only 3/9 (33.3%) of them. As a maintenance therapy, RTX was administrated at a dose of 500 mg with a 6-month interval. The remaining half of the patients (7/16, 43.8%), not receiving SSI, underwent low doses of GCs in 4/7 (57.1%), with or without IVIG, or were free from treatment in 3/7 (42.9%).

During the follow-up (median time of follow-up: 29.5 months; 25–75% interquartile range: 15.75–60 months) no serious side effects were recorded.

Relapses were infrequent during maintenance therapy. Only three relapses were noticed in three different patients (18.8%). These three patients were 75, 83 and 71 years old, and relapsed at 18th, 36th and 35th month of follow-up, respectively. All patients had been treated with triple induction therapy. The disease showed the same clinical characteristics as at the onset. Triple therapy had been suspended in two of them, and the relapse occurred after 9 and 12 months of suspension, respectively. For them, the same triple therapy as previously employed was effective again. The third relapsed patient, who was continuing the triple therapy, was successfully shifted to RTX. More details on this subgroup of patients have been reported in the Appendix A).

As required by the eligibility criteria, the anti-HMGCR antibody was positive in all the patients at baseline (Table 1). In 6/16 (37.5%) patients, the anti-HMGCR antibody titer was regularly checked during the follow-up, highlighting its progressive decrease during the treatment (Figure 2). Nevertheless, the anti-HMGCR antibody did not become negative in any of them. In one relapsing patient, for whom the anti-HMGCR antibody was available in the follow-up, after an initial and progressive decrease, the anti-HMGCR antibody titer increased three months before the increase in CK level, and showed a peak at the time of the clinical and laboratory relapse at month +18 (Figure 3). In a second relapsing patient, the increase in the anti-HMGCR antibody titer was recorded at the time of clinical worsening.

At the last clinical evaluation, all the patients were in clinical remission. The patients were either not taking GCs (9/16 (56.3%)), or were taking low doses of GCs (7/16 (43.7%)). Four patients (4/9, 44.4%) had stopped GCs within 6 months. Notably, most of the patients who discontinued GCs (8/9 (88.9%)) had been treated with triple therapy. The details about maintenance therapy are depicted in Figure 1.

## 4. Discussion

Anti-HMGCR myopathy is a rare and serious myopathy [7]. It usually affects older people during statin treatment [17]. In previous studies, type 2 diabetes and atorvastatin seemed to be associated with a higher risk of developing anti-HMGCR myopathy [4]. Anti-HMGCR myopathy can also develop in juvenile patients or in no-statin-exposed patients [18], but probably through a different genetic background [19]. Interestingly, we included one patient who developed anti-HMGCR myopathy after exposure to red yeast rice (*Monascus purpureus*), which contains mevalonic acid and Monacolin K, an agent that has the same structure and activity of lovastatin [20]. The patient recovered without treatment. The role of anti-HMGCR antibody as a discriminating factor between self-resolved toxic myopathy and anti-HMGCR myopathy has been suggested [21,22]. Nevertheless, patients with anti-HMGCR myopathy have rarely been reported to improve without SSI. [14]. Given the lack of a comprehensive description of the disease, there is no uniform approach to its management, and no target treatment recommendations [13,18,23,24].

From the first description of anti-HMGCR myopathy [7], IVIG appeared to be very useful in this disease. Nowadays, treatments are mostly based on GCs, steroid-sparing SSI, and IVIG. RTX is considered as a possible second-line therapy in case of relapse or refractoriness to a first-line treatment, due to the putative role of the anti-HMGCR antibody in the pathogenesis [10]. In our study, all the patients achieved sustained clinical remission: CK levels became normal, prior strength was restored, and, importantly, no GCs or low-dose GCs were used as maintenance therapy. These results may be considered key goals of treatment [13]. In our cohort, the triple induction therapy with GCs, IVIG and MTX proved to be both effective and safe, despite the old age and the co-morbidities of the patients (Table 2). Moreover, relapses were infrequent during the maintenance therapy. The maintenance therapy was based on SSI with or without IVIG and GCs, aiming to tailor it to each patient. From a preliminary study, the predictive factors for a successful maintenance with SSI monotherapy are considered IVIG as induction therapy and avoidance of diagnostic-therapeutic delay [13].

Our observations are in line with the results reported by Meyer et al. from Canada [13]. In fact, Meyer et al. described a retrospective cohort of 55 patients, most of them treated with triple induction therapy, and a quarter were treated with a GC-free induction strategy. Currently, GC-free induction therapy and GC-free maintenance are interesting strategies in anti-HMGCR myopathy treatment. The opportunity to treat anti-HMGCR myopathy without GCs [13] could be a key issue in the context of such elderly patients presenting with cardiovascular co-morbidity and diabetes. In our cohort of patients, GCs were usually employed in induction therapy, but high (i.e., 0.5 mg/kg/day or more) or medium doses (less than 0.5 mg/kg/day but higher than 15 mg/day of prednisone-equivalent) of GCs were used only for a short time (i.e., the first three months) in most patients (62.5%). Otherwise, IVIG and MTX were used very frequently also to allow a quicker GC tapering. Notably, in our study, MTX was often employed in the induction therapy and was then continued in the maintenance phase. While the usefulness of MTX in induction therapy, in addition to IVIG, remains to be demonstrated, an early treatment with MTX may make the clinician more confident in reducing GCs and, also, the overall duration of IVIG treatment. The usefulness of MTX is well documented in IIM, as well as in elderly patients [25]. However, further studies are needed to address this issue.

Thus, our study supports the usefulness of SSI and IVIG therapy, with the opportunity to employ lower cumulative dose of GCs. Overall, our results reinforce the concept that anti-HMGCR myopathy may be not a GC-dependent disease (Figure 1). A key point is the early diagnosis of anti-HMGCR myositis. First, patients with normal strength (but increased CK levels) at diagnosis may be successful candidates for GC-free induction [13], and secondly, a deeper knowledge of the disease, its onset and progression would be relevant to better stratify patients according to disease severity, in order to support the choice of a low-GC or even GC-free regimen treatment. Currently, the age at onset has been suggested as a possible criterion for a more aggressive or less intense immunosuppression, since patients above the age of 60 could suffer from a less severe disease and show a better outcome [14]. In our cohort there was also a similar trend between CK and the anti-HMGCR antibody, as already published [14]. Indeed, the observation that anti-HMGCR antibody titer increased just before the clinical and laboratory relapse in one patient points to the opportunity to explore this antibody as biomarker of relapse, and the successful response to RTX herein reported in one relapsed patient, might further stimulate the research on the putative role of the B cells and the anti-HMGCR antibody in the pathogenesis of the disease [10,11,26].

## 5. Strengths and Limitations

This study has some limitations. The cohort is small and the patients were collected from different specialized centers in a wide range of time; however, the disease is indeed very rare, while the treatment strategy was quite similar in all the recruiting centers, despite the different clinical backgrounds. The design of the study is retrospective, but the rarity of the disease makes it difficult to plan randomized controlled trials with an adequate statistical power and a reasonable duration. Notably, it should be taken into account in the interpretation of the results that the good outcome we documented in our cohort may be also related to age. In fact, 13 out of 16 patients were older than 60 years, and it has been suggested that younger patients have a more severe muscle disease than older patients and a worse prognosis [14]. The analysis of the anti-HMGCR antibody titer was determined in a few of the patients during the follow-up and only in one who showed a relapse. Since the anti-HMGCR antibody might be a prognostic biomarker, this analysis needs to be thoroughly investigated. The muscle biopsy was performed in 10 out of 16 patients. While muscle biopsy could not be required for the diagnosis of anti-HMGCR myopathy, as the specificity of the antibody is very high [27], the histopathology has been considered relevant for the prognosis by the European Neuro Muscular Centre (ENMC), since the presence of muscular dystrophy or perifascicular atrophy is useful to better differentiate and subclassify patients with this type of myositis [28]. Atrophy was found in only one patient among our biopsied patients. This feature could have contributed to the favorable outcome of our cohort, even if patients with refractory disease did not show coexisisting muscular dystrophy, as in Tiniakou et al. [14].

## 6. Conclusions

The anti-HMGCR myopathy is a severe but treatable disease characterized by the presence of anti-HMGCR autoantibody, elevated serum CK levels, and proximal skeletal muscle weakness.

The triple- or double-IVIG-based therapy is an effective and safe strategy; it may allow low cumulative doses of GCs and even a steroid-free maintenance. Further studies are required to personalize the treatment, in particular, in this context of often frail patients.

## Figures and Tables

**Figure 1 jcm-09-03454-f001:**
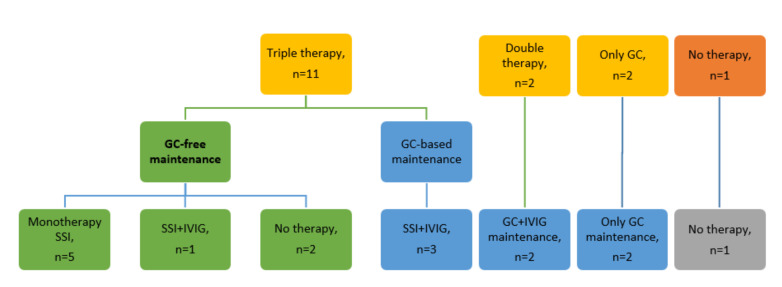
Flow diagram of induction and maintenance therapy. Yellow squares represent the induction therapy; the green and blue ones represent the GC-free and GC-based maintenance, respectively. Legend: GC, glucocorticoids; IVIG, immunoglobulins; SSI, steroid-sparing immunosuppressant.

**Figure 2 jcm-09-03454-f002:**
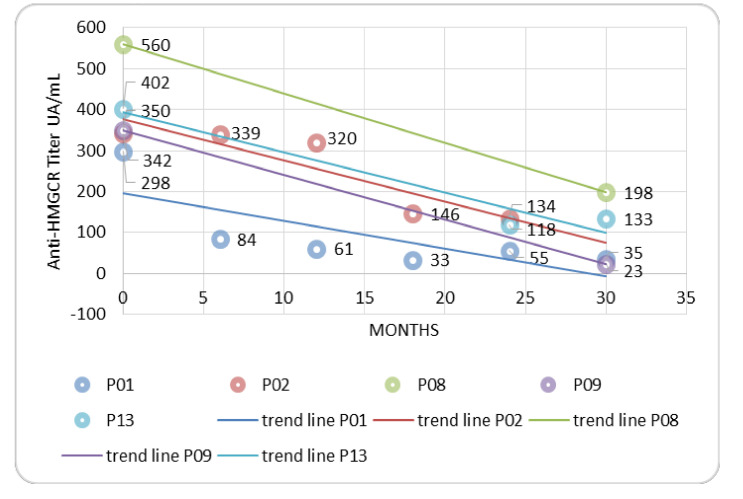
Anti-HMGCR antibody in patients who achieved sustained remission of disease. Legend: anti-HMGCR, anti-hydroxy-methylglutaryl coenzyme A reductase.

**Figure 3 jcm-09-03454-f003:**
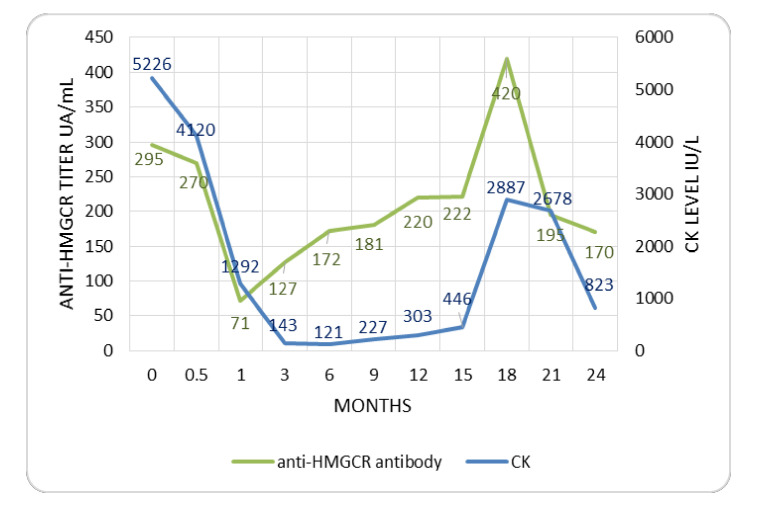
Anti-HMGCR antibody titer and CK level in a relapsing patient. Legend: CK, creatine kinase; anti-HMGCR, anti-hydroxy-methylglutaryl coenzyme A reductase.

**Table 1 jcm-09-03454-t001:** Demographic features, clinical presentation, laboratory parameters, management and outcome of patients with anti-HMGCR myopathy (*n* = 16).

Pts	Age at Onset/ Gender/ Ethnicity	Statin Exposure and Drug	Type of Muscle Weakness (MRC)	CK (IU/L)	Anti HMGCR Antibody	Biopsy Findings	Induction Therapy	Time to Achieve Clinical Remission	Time to Reach Low-Doses of GCs (<15mg/day Prednisone-Equivalent)	Outcome (Maintenance Therapy at Last Follow Up)	Relapse
**P01**	76/F/ Caucasian	Yes, atorvastatin	Upper and lower limbs (3/5), myalgia, skin involvement	13171 IU/L	Positive, 289 UA/mL	Lymphocytic infiltrates and polyfragmentation of elastic fibers	Steroid, IVIG, MTX	6 months	2 months	Remission, (MTX)	No
**P02**	78/M/ Caucasian	Yes, atorvastatin	Upper and lower limbs (3/5), myalgia, dysphagia	6181 IU/L	Positive, 320 UA/mL	Necrosis, atrophic fibres, secondary mitochondrial alterations, absent inflammatory infiltrate	Steroid, IVIG, MTX	22 months	3 months	Remission (MTX)	No
**P03**	74/M/ Caucasian	Yes, atorvastatin	Upper and lower limbs (3/5), myalgia	5226 IU/L	Positive, 295 UA/mL	Biopsy not performed	Steroid, IVIG, MTX	3 months	1 months	Remission (MTX, IVIG), one relapse	Yes
**P04**	72/M/ Caucasian	Yes, atorvastatin	Lower limbs (3/5), myalgia	5649 IU/L	Positive, 192 UA/mL	Biopsy not performed	Steroid, IVIG, MTX	6 months	2 months	Remission (MTX)	
**P05**	80/F/ Caucasian	Yes, atorvastatin	Upper and lower limbs (3/5), myalgia, dysphagia, skin involvement	3585 IU/L	Positive, NA	Minimum focal inflammation, CD3+/CD8+, CD68+ cells	Steroid, IVIG, MTX	3 months	3 months	Remission (MTX, IVIG, GCs)	No
**P06**	66/M/ Caucasian	Yes, atorvastatin	Upper and lower limbs (1/5), dyspnea with oxygen therapy, dysphagia with parenteral nutrition	11393 IU/L	Positive, NA	Necrosis	Steroid, IVIG, MTX	3 months	3 months	Remission(IVIG), one relapse	Yes
**P07**	80/F/ Caucasian	Yes, atorvastatin	Upper and lower limbs (2/5)	3540 IU/L	Positive, NA	Muscle fiber phagocytosis	Steroid, IVIG	2 months	4 months	Remission (IVIG, GCs)	No
**P08**	49/F/ Caucasian	No	Upper and lower limbs (3/5), dysphagia	359 IU/L	Positive, NA	Necrosis, myopathic alteration without inflammatory deposits	Steroid, IVIG, MTX	6 months	6 months	Remission (AZA, IVIG, GCs)	No
**P09**	76/M/ Caucasian	Yes, atorvastatin	Upper and lower limbs (2/5), myalgia, skin involvement	5733 IU/L	Positive, 350 UA/mL	Necrosis, inflammation	Steroid, IVIG, MTX	12 months	3 months	Remission (no therapy)	No
**P10**	86/M/ Caucasian	NA	Upper and lower limbs (3/5), myalgia	2100 IU/L	Positive, 62 UA/mL	Biopsy not performed	Steroid	9 months	6 months	Remission (GCs)	No
**P11**	82/M/ Caucasian	Yes, atorvastatin	Upper and lower limbs (3/5), dysphagia	9600 IU/L	Positive, NA	Biopsy not performed	Steroid, IVIG, MTX	9 months	5 months	Remission (MTX)	No
**P12**	55/F/ Caucasian	Yes, red rice	Lower limbs (4/5), Myalgia	1000 IU/L	Positive, NA	Biopsy not performed	No therapy, suspension red rice	3 months	Non applicable	Remission (no therapy)	No
**P13**	68/F/ Caucasian	Yes, atorvastatin	Upper and lower limbs (1/5), myalgia, dysphagia	10345 IU/L	Positive, NA	Biopsy not performed	Steroid, IVIG, MTX	6 months	3 months	Remission (RTX), one relapse	Yes
**P14**	81/M/ Caucasian	NA	Upper and lower limbs (3/5), dyspnea without oxygen therapy, acute renal failure	1377 IU/L	Positive, NA	Necrosis	Steroid	18 months	2 months	Remission (GCs)	No
**P15**	60/M/ Caucasian	Yes, atorvastatin	Upper and lower limbs (4/5), myalgia	8900 IU/L	Positive, 161 UA/mL	Necrosis	Steroid, IVIG, MTX	9 months	6 months	Remission (IVIG, MTX, GCs)	No
**P16**	75/F/ Caucasian	Yes, atorvastatin	Upper and lower limbs (4/5)	7000 IU/L	Positive, 126 UA/mL	Necrosis	Steroid, IVIG	6 months	5 months	Remission (IVIG, GCs)	No

Legend: anti-HMGCR, anti-hydroxy-methylglutaryl coenzyme A reductase; AZA, azathioprine; CK, creatine kinase; EMG, electromyography; GCs, glucocorticoids; IVIG, immunoglobulins; MRC, Medical Research Council; MTX, methotrexate; Pts, patients; RTX, rituximab.

**Table 2 jcm-09-03454-t002:** Baseline characteristics of the patients with anti-HMGCR myopathy (*n* = 16).

**Demography**	**N (%); mean ± SD or Median (Range)**
Age, years	72.4 ± 10.3
Female gender	7 (43.8)
Prior statin use	13 (81.3)
Atorvastatin	12/13 (92.3)
Type 2 diabetes	7 (43.8)
Cardiovascular disease	8 (50)
Arterial hypertension	11 (68.8)
Cancer during follow-up	0 (0)
**Baseline Muscle Evaluation**	
Proximal muscle weakness	15 (93.8)
MRC score	3 (1–4)
Myalgia	10 (62.5)
Oropharyngeal dysphagia	7 (43.8)
Serum CK level at onset (IU/L)	5691 (359–13171)
Muscle biopsy availability	10 (62.5)
Necrosis	9/10 (90)
Myopathic EMG	14/14 (87.5)

Legend: SD, standard deviation; CK, creatine kinase; EMG, electromyography; MRC, Medical Research Council.

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
