# Peer review of "Efficacy and Safety of High-Dose Immunoglobulin-Based Regimen in Statin-Associated Autoimmune Myopathy: A Multi-Center and Multi-Disciplinary Retrospective Study"

_jcm, 2020, doi:10.3390/jcm9113454_

Round 1

Reviewer 1 Report

Response to “Efficacy and safety of high-dose immunoglobulin-based regimen in statin-associated autoimmune myopathy: a multi-center and multi-disciplinary retrospective study”

There is a need for further understanding of the clinical subsets of patients with suspected statin-induced myopathy to improve diagnostics, the clinical care and in the end the prognosis of these patients.

This paper describes the treatment and disease course of 16 patients with anti-HMGCR positive immune-mediated necrotizing myopathy.

I have some major concerns about the study. The abstract and introduction must improve. In addition, the study is lacking crucial information about the patients and I don´t find information on Ethical approval of the study. I have organized my major and minor comments and questions below by pages of the manuscript.

Major comments:

  1. Page 1, Abstract; please follow the author guidelines. The abstract has a very long background section, appears disarranged and without mentioning methods.
  2. Author guidelines: 1) Background: Place the question addressed in a broad context and highlight the purpose of the study; 2) Methods: Describe briefly the main methods or treatments applied. Include any relevant preregistration numbers, and species and strains of any animals used. 3) Results: Summarize the article's main findings; and 4) Conclusion: Indicate the main conclusions or interpretations.
  3. Page 2, introduction; please follow the author guidelines; “The current state of the research field should be reviewed carefully” which is indeed very short in this paper. “Finally, briefly mention the main aim of the work and highlight the main conclusions.” however the findings/conclusions section within the introduction of this study is very long.
  4. Page 2, line 87;
    1. Ethical approval? Which is mandatory
    2. Registered at The National Data Protection Agency?
  5. Page 3, line 132; How was relapse defined? Raised CK, complains of decreased strength by the patients or objectively by muscle test revealing decreased strength? Please specify remission/relapse in the method section as it´s of major importance to this paper. It might be what you put in the discussion section, line 171: " all the patients achieved sustained clinical remission: CK levels became normal, prior strength was restored, and, importantly, no GCs or low dose GCs were used as maintenance therapy.” If muscle tests were not included as a measure of disease activity and remission, it should be mentioned as a limitation in the discussion section.

Minor comments:

  1. Page 2, line 80; “clinical and laboratory evidence of muscle injury (i.e. elevation of serum CK levels). More details please.
  2. Page 2, line 81; Definition of “positive anti-HMGCR”, cut-off value?
  3. Page 2, line 96; diagnosed between 2012-2019 and not collected between 2012-2019? How was the selection actually done at each center? Did they run through al their myositis patients diagnosed between 2012-2019 and identified these patients with positive anti-HMGCR or how? Was the selection done in the same way in every center?
  4. Page 2, line102; “The median serum CK level at the onset of the disease was 5649 IU/L”, what was the reference interval, including the upper limit of CK?
  5. Page 2, line 106; “The muscle biopsy was performed in 10/16 (62.5%) patients”, that´s a limitation, which should be mentioned in the discussion section.
  6. Page 3, line 113; “Two out 16”, you miss “of”
  7. Page 3, line 116; “Finally, the only patient exposed to red rice resolved with red rice suspension only.” Should this patient be included in the study at all? Wasn´t it just a case of toxic myopathy? I´m not convinced of the diagnosis immune-mediated necrotizing myopathy.
  8. Page 3, line 111: Please refer to Fig 3 in the beginning of the treatment and outcome section
  9. I strongly recommend a table with details of each patient including: age/gender/clin and lab data of myopathy/treatment/follow-up data
  10. Page 4, line 144; “Anti-HMGCR antibody * in patients who achieved sustained remission of disease.” Don´t you miss *“titer” in the legend? In addition, I would recommend moving the numbers in the figure 1 and 2, which you don´t really need.
  11. Page 5, line 165; “as there is no activity/severity score”. I would strongly recommend refrasing as we have the recommendations from IMACS; https://www.niehs.nih.gov/research/resources/imacs/diseaseactivity/index.cfm
  12. Page 6, line 171; “prior strength was restored”. What was this assumption based on? I don´t find any references to muscle tests in the method or result section.

Author Response

Many thanks to the reviewer for the valued suggestions which have substantially improved the manuscript, in our opinion.

Please see the attachment for point to point answers.

Reviewer 2 Report

The authors present a case-series of 16 patients with anti-HMGCR myositis, who were followed in 6 different centers in Italy. The majority of the patients were treated with some combination with IVIG, and all of them improved during the time frame they were followed. This is a well-written case series, although some of the findings have been described in larger cohorts and have answered the question if anti-HMGCR titers can be used as a biomarker. However, more attention could be given to other elements, like the patients that relapsed.

  1. How high was the titer for the patient on no treatment? What kind of symptoms were present and what improved after discontinuation of red yeast?
  2. Line 52: the antibodies have been implicated in the disease pathogenesis, but would not say recognized as pathogenic (there are many clinical arguments against it, and some considerations regarding experimental design)
  3. Line 172: this is not described in the results section
  4. All patients were above the age of 60, so they have better prognosis and that could explain the clinical remission. According to a study by Hopkins (Tiniakou et al, Rheumatology 2017), 75% of patients above the age of 60 will recover their clinical strength within 2 years.
  5. The observation that CK and anti-HMGCR titers correlate is not new, and it is described at the same paper, where they followed CPK and anti-HMGCR titers of 50 patients followed for at least 2 years. The improvement of the CPK/anti-HMGCR was inversely correlated with the increase in muscle strength. Therefore, one could argue that CPK is a better marker, given that it is readily available.
  6. It would be interesting to further describe the three patients that relapsed off treatment; for example, how long was the initial treatment and how long after discontinuation, did they exhibit first signs of relapse.

  1. SSI is not defined when it is first used.
  2. Figure 2-maybe can use different colors for the numbers to better distinguish CPK from antibody titer
  3. Please define low-moderate-high dose GC
  4. Line 187: Quantification of “short time”

Author Response

(The authors gave the same response as above.)

Round 2

Reviewer 1 Report

I find that the revised manuscript has improved significantly and I have only one concern left however of mandatory. I still don´t find any Ethical approval for using clinical data for research purpose, which is mandatory.

Author Response

Dear Reviewer,

thanks for the appreciation of the new version of our manuscript.

We better clarified the compliance with ethical standards in the Paragraph 2.4, page 3, lines 107-112.

Sincerely,

Luca Quartuccio, M.D., Ph.D.

Response to Reviewer 1:

The paragraph 2.4 now reports: The retrospective study was approved by the Local Institutional Review Boards (IRB) from all the centres involved (University of Udine IRB, coordinating centre, prot. n. 79577). The off-label use of IVIG for anti-HMGCR myopathy was approved by each hospitals. All the patients were asked to sign an informed consent for accessing their medical records and using their clinical data for research purposes and improvement of healthcare.

Reviewer 2 Report

The authors have done a very good job editing the manuscript and it flows much better. My only comment is if they can include a sentence about red rice yeast, and why it might be a potential trigger.

Author Response

Dear Reviewer,

thanks for your appreciation of the new version.

We enclosed a sentence about red yeast rice and the mechanism of myopathy of it in the Discussion (Page 3, Line 201-202). A reference (mini-review) to this issue has been added (Farkouh A, Baumgärtel C. Mini-review: medication safety of red yeast rice products. Int J Gen Med 2019;12:167-171).

Sincerely,

Luca Quartuccio, M.D., Ph.D.

Response to Reviewer 2:

The new sentence states that: ”Interestingly, we included one patient who developed anti-HMGCR myopathy after the exposure to red yeast rice (Monascus purpureus), which contains mevalonic acid and Monacolin K, an agent that has the same structure and activity of lovastatin[20]”.